# How Do Hunters Hunt Wild Boar? Survey on Wild Boar Hunting Methods in the Federal State of Lower Saxony

**DOI:** 10.3390/ani11092658

**Published:** 2021-09-10

**Authors:** Oliver Keuling, Egbert Strauß, Ursula Siebert

**Affiliations:** 1Institute for Terrestrial and Aquatic Wildlife Research, University of Veterinary Medicine Hannover, Bischofsholer Damm 15, 30173 Hannover, Germany; egbert.strauss@tiho-hannover.de (E.S.); ursula.siebert@tiho-hannover.de (U.S.); 2Hunting Association of Lower Saxony, Landesjägerschaft Niedersachsen e.V., 30625 Hannover, Germany

**Keywords:** human dimension, hunting methods, monitoring, private hunting, *Sus scrofa*, wild boar, wildlife management

## Abstract

**Simple Summary:**

High wild boar population densities lead to human–wildlife conflicts. For proper wildlife management, knowledge of wildlife biology as well as human attitudes is needed. We conducted inquiries on hunting methods and on hunters’ attitudes in the German Federal State of Lower Saxony to better understand hunting strategies. Single hunt, especially at bait, is still the most widely used method for hunting wild boar. The proportion of drive hunts within the hunting bag is increasing. The proportions of hunting methods vary regionally due to wild boar densities, geographical conditions and hunters’ practices. Private hunting is important for wild boar management, although it is just insufficient. Besides promoting more efficient hunting methods and motivating hunters, in the future, additionally, administrative wildlife managers could be established as coordinators of wild boar management, and as such, could manage hunting, the incorporation of regional conditions and investigating hunters’ attitudes and abilities.

**Abstract:**

High wild boar population densities lead to demands for a population reduction to avoid crop damages or epidemic diseases. Along with biological studies, a better understanding of the human influence on wildlife and on wildlife management is important. We conducted inquiries on hunting methods and on hunters’ attitudes in the Federal State of Lower Saxony, Germany, to better understand hunting strategies and the influence on increasing wild boar population, as well as to underpin game management concepts. Single hunt, especially at bait, is still the most widely used method for hunting wild boar. The proportion of drive hunts within the hunting bag is increasing. The proportions of hunting methods vary regionally due to wild boar densities, geographical features (vegetation, terrain, etc.) and hunters’ practices. Hunters increased the proportion of conjoint hunts on wild boar. Baiting remains an important hunting method in wild boar management and the proportion of drive hunts should be fostered. Private hunting is important for wild boar management, although it is just insufficient. Additionally, administrative wildlife managers are recommended for the near future as coordinators of wild boar management, and as such, could manage hunting, the incorporation of regional conditions and investigating hunters’ attitudes and abilities.

## 1. Introduction

Wild boar *Sus scrofa* L. hunting bags are at a very high level in Germany and all over Europe [1,2]; populations are still increasing and dispersing into agrarian landscapes. Thus, high densities and dispersal into new areas are accompanied by economic problems such as crop damages and disease transmission to livestock, threats to nature conservancy as well as threats to human wellbeing, e.g., by car accidents [3,4,5,6,7,8]. Due to factors such as insufficient hunting, as well as underestimation of population densities and reproductive rates [9], harvest rates seem to be insufficient in many parts of the wild boar range [1]. Sound hunting management is crucial in order to regulate or even reduce wild boar populations [1,10,11,12]. However, it seems hunting alone is not able to regulate wild boar populations sufficiently [1,2,12,13]. In that context, it seems necessary to have only a few unmanaged areas (compare [14]).

Throughout the past five decades, wild boar research in Europe [15,16,17,18] has lacked attention to certain biological aspects [18], e.g., habitat use [19], dispersal mechanisms [20,21], group structure and social hierarchy [22], which require examination. In addition, the human factors directly or indirectly influencing wildlife—the so-called “human dimension”—is a relatively new topic in Europe, especially in Germany [6,23,24,25,26,27], and necessitates further investigation [6,12,23,24,25,28,29,30].

It is generally assumed that biased sex and age ratios, in addition to high hunting pressure (e.g., [31,32]), result in increased reproduction, although food availability and nutritional content were demonstrated as the main cause for higher reproduction [9,33,34]. Some authors described different models to accomplish regulation of wild boar populations by hunting different proportions of age classes [21,35,36,37,38]. Although information on wild boar or feral pig management does exist, it comes mainly from Australia and the USA [39,40,41,42], only sketchy from Europe [43,44,45,46,47,48]. The scientific literature on hunting practices in Europe is vague in general [49] with some on single hunts [50], a bit on drive hunts that focusses mainly on a biological view (activities, behavioural changes; [51,52,53,54,55,56]), and fewer on efficiency [2,51,57,58,59] or effort [60,61]. Additionally, in other regions, the literature on hunting techniques and efficiency is sparse [62,63,64,65].

In order to counteract economic problems such as damage to crops, disease outbreaks, and traffic accidents, the regulation of the wild boar population must change. For sufficient population regulation, and especially for the population reduction necessary to halt infection chains and reduce crop damages [1,66], the concept of “hunting management” should be expanded into a comprehensive “wildlife management” concept. This new form of management would oversee everything from private and recreational hunting, along with other anthropogenic influences on wildlife [2,12], to nature conservation. Once this is accomplished, it may be expected that regulation of the population of wild boar may be possible, as long as both public awareness and impact are substantial enough [67], and additional management measures might be accomplished by professionals. Some significant requirements are necessary for the success of management programs [12,24]:Evaluating the opinion and attitudes of stakeholders, hunters, and the general public;Incorporating scientific background, social needs, and social attitude;Monitoring the success of management measures.

Of course, involved people will have to agree with the management plans.

In order to regulate the hunting of wild boar, it is essential to research the animal’s biology (especially behavioural and ecological flexibility; [68]), hunting methods, and the attitudes and availabilities of hunters. One of the most significant and disturbing factors affecting the success of wildlife management is the management itself: wildlife managers, and especially the hunters, have a big impact on the performance of population regulation [1,2,12,27,69,70,71]. For example, is the number of hunters linked to the hunting bags of wild boar [2]? Also, it is important to determine a status quo in order to develop management, i.e., how is management (by private hunting) actually conducted?

Hunting bag data is influenced by several factors including weather conditions, nutrition, hunting methods, population densities and visibilities, and only reflect approximate/trend-based population data. Due to several EU conventions and German laws, reliable population data is needed [72,73]. Thus, wildlife monitoring systems, specifically for game species, were installed in Germany and its federal states in the last three decades [72,74]. In the Federal State of Lower Saxony monitoring systems that track distribution, detailed hunting statistics, as well as the spectrum of opinion of the hunters’, work on a volunteer questionnaire basis and can be used for human dimension surveys. Besides inquiries on hunters’ opinion, knowledge, abilities and willingness, we also conducted a survey on hunting itself. This knowledge should be used when considering management concepts and subsequent decisions, or even in making recommendations to hunters on how they could improve hunting efficiency. We aimed to know: How do hunters hunt? Are hunters willing (compare [27]) and able to regulate a population sufficiently? What other management tools can be used and accepted by hunters? Therefore, we asked the owners of hunting grounds how hunting was conducted throughout the last two decades with a specific view on differences between regions and trends in hunting techniques.

## 2. Materials and Methods

### 2.1. Study Area: Lower Saxony

The Federal State of Lower Saxony is situated in north-western Germany and is with 47,624 km^2^ the second-largest federal state of Germany. Lower Saxony borders the Netherlands to the West, the North Sea in the northwest and nine other federal states in the northeast, east and south (see Figure 1).

The state’s administration is structured into 427 municipalities which form 46 districts. The human population density is 160 inhabitants/km^2^. In this region, the human population’s moods and opinions are as diverse as the variety of landscapes which allows also for comparison with other human populations in other regions. The landscape is formed by several different macrochores (main physical-geographic regions, Figure 1). Three-quarters of the area are formed by the North German Lowlands with natural sceneries such as the Wadden Sea and marshes in the north; intensive agrarian landscapes with crop fields and grassland, some heath, moor and bushlands, as well as smaller forests on flatlands with gentle hillocks, make up the northern and western parts (containing the macrochores: N northern lowlands, NW north-western lowlands, SW south-western lowlands and hills). The mean hunting index (HI = dead individuals/km^2^) in this sector was <1. The centre is made up of strongly agriculturally-dominated areas (CL central lowlands), as well as more forested areas on the gentle hills of the Luneburg Heath (NE north-eastern lowlands) in the east (HI > 1–3). The remaining quarter (HI > 1–4) is formed by forested areas, also containing agriculture, on the low mountain ranges (S southern hills and low mountain ranges) up to the Harz (H) Mountains in the south (Wurmberg 971 m asl) (in the following mainly combined as S + H). Agriculture forms 60% of the area of Lower Saxony, 22% is forested, 2% is fresh water, and 16% are of anthropogenic origin (settlement, industry, traffic, etc.; no hunting allowed). A total of 29% of the forested areas (7% of the total area) are managed by the Forestry Offices of Lower Saxony, which contribute about 15% of the annual hunting bag of Lower Saxony. The climate is temperate on the transition zone between the Atlantic in the northwest, with a continental climate in the east. The average annual temperature is around 8 °C. The rainfall ranges from 500 mm per year in the east to 800–900 mm in the northwest and up to 1000–1600 mm at the western slopes of the Harz Mountains.

In the Federal State of Lower Saxony, as in total Germany, the hunting regime is organised in hunting grounds. The area of private hunting grounds in Lower Saxony ranges between 75 ha (minimum legal area) to more than 1400 ha and has a mean of about 500 ha. Owners or tenants of the hunting grounds hunt in personal responsibility with a compulsory duty for wildlife management (“Pflicht zur Hege”; duty for population regulation, custody and conservation; compare also Appendix A). It is important to distinguish between private hunters (own responsibility for hunting ground and game), professional hunters (employed professional hunters or forestry officers, responsible for common or private hunting grounds, professionally educated) and some few recreational hunters (using a kind of license hunting system without any responsibilities). Hunting solely is allowed with the permission of the hunting ground owners. Thus, in the following we deal with the opinion of hunting ground owners or tenants, calling those “private hunters”. In Germany, no additional management of game species by official gamekeepers (such as the “Wildhut” in Switzerland) is conducted. Recently, wild boar hunting in Lower Saxony is allowed all year round. Solely the shooting of sows leading striped piglets is forbidden [75]. There are no quotas, as hunters are supposed to hunt as many wild boars as possible. Baiting still plays a very important role [57,66,76]. There are only very few areas in Lower Saxony, where game species are unmanaged (hunting ban in core zones of national parks and some few nature conservancy areas, total area unknown).

For more details on the German hunting regime see Appendix A. Source of general data in section “Study area”: climate data: DWD [77]; hunting bag data are based on official hunting bag statistics from NMELV [78]; other data: Statistik-Portal [79].

### 2.2. Data Collection

The wildlife survey in Lower Saxony (WTE = Wildtiererfassung in Niedersachsen) was established in 1991 on behalf of the Hunting Association of Lower Saxony (Landesjägerschaft Niedersachsen e.V., funded by hunting charges–Ministry of Agriculture) initially as a small game monitoring system. From the start, the WTE was conducted in cooperation with the Institute for Wildlife Research and its successor, the Institute for Terrestrial and Aquatic Wildlife Research (ITAW), University of Veterinary Medicine Hannover, Foundation [73].

Within the WTE we conduct inquiries on several game species, mainly on small game occurrences [72,73,74]. Along with simple questions on occurrences or abundance/density estimates of several game species, hunters are asked about hunting bags, diseases, road kills, crop damages, and other topics including the human dimension, e.g., hunters’ opinions, attitudes, abilities, and hunting methods in their own hunting grounds. The inquiries are sent to every tenant or owner of a private hunting ground via the hierarchical structures of the hunting association. The state’s forestry offices are also involved in the WTE, but for organisational reasons (the forestry offices participate in the general monitoring, but not in opinion polls, as they conduct their own internal company surveys) are not incorporated in this study. The annual response rate is 85–90% from about 9100 hunting grounds. While this may not reflect the true opinions of all 60,000 hunters in Lower Saxony, all hunters are required to abide by the terms set by the tenants or owners of the hunting grounds, regardless of their individual views. Thus, the results of this survey reflect how hunting is actually conducted area-wide. In the following study, we will mostly call the respondents (owners and tenants of the hunting grounds or their “adjutants”) simply “hunters”. Presented are the total number of answers (N = number of responding hunting grounds). The reliability of population estimates within WTE is frequently evaluated and calibrated by different census methods [73,74]. Data are analysed on the municipality level or several regional levels (e.g., macrochores, districts).

The data for this study were collected during the years 2002–2020 with a detailed interrogation on hunting methods in the years 2010–2012 (compare Appendix A). For different comparisons we used the combined data of the above-mentioned macrochores N, NW, SW, NE, CL, S + H, and did a simple summary of the data of west (W) and east (E) Lower Saxony (see Figure 1) or summarised the data of different wild boar frequencies. To prove differences of percentages between categories we calculated Chi-square goodness of fit tests for nominal data (yes/no) and a Kruskal–Wallis test for number of baiting stations. All simple statistics proceeded with their own formulas in MS^©^ Excel. To characterise the development of proportions over time we calculated linear regressions determining whether the slopes of tendencies were negative or positive. Regression was carried out by fitting the time series of the various hunting methods via linear models (lm) in R [80]. The overall significance level for all tests and models was alpha = 5%.

Hunters always had the option to give a “no”—answer in the form of “I do not want to answer”. The specific questions will be presented in the results section.

## 3. Results

Most questions were answered with the usual response rate of 85–90% (from 7285–8059 participating hunting grounds in 2002–2020—representing hunting years 2001/02–2019/20). Only some specific questions on hunting methods and the objectives leading to this were only answered by fewer hunters, mainly depending on the occurrence of wild boar more or less regularly [compare [26]]. In addition, we think some of these questions had imprecise wording or were too private/too blunt. To avoid giving the wrong impressions, we chose not to analyse this data further.

### 3.1. Hunting Methods + Hunting Bags

#### How Many Wild Boar Were Shot in Your Own Hunting Ground during the Last Season with Which Hunting Method?

Predetermined answers: single hunt, drive hunt within own hunting ground, conjoint drive hunt—meaning hunts on several neighbouring hunting grounds at the same time (see Figure 2, more specified in the years 2010 and 2011 see Figure 3). For an explanation of hunting methods see Appendix A.

Hunting at bait was the most common hunting method in Lower Saxony in the years 2009/10 and 2010/11 (Figure 3) with a recently decreasing proportion (compare Table 1, Figure 2). The proportions of hunting methods within hunting statistics differ based on region (Table 2, Figure 3). Since 2005, the proportion of wild boar shot on drive hunts has increased (Figure 2, Table 1). After an increase in wild boar shot on drive hunts within the own hunting ground until 2010 this proportion was lowered again since 2011 and showed just a slight decrease during the last two decades (thus, there was no change detectable; Table 1), whilst the proportion of wild boar shot on conjoint drive hunts was increasing continuously (Figure 2, Table 1). Meanwhile, the proportion of wild boars on a single hunt did decrease (Figure 2, Table 1).

### 3.2. How to Hunt?

#### 3.2.1. Do You Bait Wild Boar in Your Hunting Ground?

(N = 7429, database: WTE 2012, singular query in 2012)

There were regional percentage differences on hunting grounds where bait was used (Figure 4). In hunting grounds in regions with higher frequencies or densities of wild boar, baiting was conducted more frequently (chi^2^ goodness of fit test: *x*^2^ = 446.6, df = 5, *p* < 0.001). Baiting was more common in regions with higher wild boar densities and less common regions with fewer wild boar occurrences. Those hunting grounds with baiting stations had a mean of 2.6 baiting stations per hunting ground; this did differ between regions (W: 2.2 bs/hg, E: 2.7 bs/hg, see Figure 1; Kruskal–Wallis test: *x*^2^ = −4541.5, df = 5, *p* < 0.001).

#### 3.2.2. Do You Conduct Drive Hunts on Wild Boar?

(N = 7907, database: WTE 2012, singular query in 2012).

The proportion of hunting grounds where drive hunts were conducted differed due to the occurrence, frequency and density of wild boar (Figure 5, chi^2^ goodness of fit test: *x*^2^ = 836.7, df = 3, *p* < 0.001). Only half of the hunting grounds in areas with resident wild boar conducted joint hunts.

#### 3.2.3. On the Drive Hunts You Conduct: Wild Boar Is (A) Main Species or (B) Incidental Species (“Bycatch”)? Which Other Game Species Are Hunted?

(N = 3234 hunting grounds conducting drive hunts at all, database: WTE 2012, singular query in 2012)

In about one-third of all private hunting grounds drive hunts were conducted. Wild boar was considered as the main or aim species in 52% of hunts and considered as an incidental species (“bycatch”) in 21% of hunts. Other ungulate species were hunted regionally and their occurrences differ as follows: red deer *Cervus elaphus* (20.6%), fallow deer *Dama dama* (21.3%), mouflon *Ovis orientalis musimon* (2.4%). On 79% of private hunting grounds, that were conducting drive hunts and answering our questions, roe deer *Capreolus capreolus*, 82% meso-predators such as red fox *Vulpes vulpes* and martens, all of which are common species, were hunted along with wild boar.

#### 3.2.4. How Do You Conduct Drive Hunts?

(a)conduction “hunting method”

(N = 1969, database: WTE 2012, predetermined answers, see Appendix A).

Specifically, 65% beat thickets, 16% drove reed and 29% drove copses (within selective drives) (only small overlap of doing at least two of those; Appendix A). Additionally, beating maize had high percentages in the northern (60%) and north-western (63%) regions. Intertillage was driven regularly in agricultural regions (33–67%), but only seldom in forested areas (11–15%). Heavy drive hunts had their focus in the eastern parts of Lower Saxony (48–59%), whilst poking (driving only with dogs, no beaters; 5–23%) and slow beating (17–44%) were conducted much less. Collective hides were conducted in about 40% of all answering hunting grounds similarly in all regions. A total of 1467 hunting grounds answered whether they hunted with (94%) or without (6%) dogs.

(b)hunting party

(N = 2224, WTE 2012, predetermined answers):

51% hunted with invited hunters (e.g., neighbours, friends), 48% hunted solely with friends and “co-hunters”, as well as 1% of private hunting grounds had guests who had to pay for the hunt. The proportions of invited hunters differed between regions (chi^2^ goodness of fit test: df = 5, *p* < 0.001, friends: *x*^2^ = 261.9; invited guest: *x*^2^ = 240.9; paying guests: *x*^2^ = 20.6).

(c)type of stands*

(N = 2119, database: WTE 2012, singular query in 2012, *stands are defined as the position and facility where the hunters are placed)

About 50% used combinations of different types of stands for drive hunts (thus, we have a total count of 3180 answers). Preferred were drive hunt stands (low raised hides ~1.5 m = 44%), followed by stands on ground level (30%), and typically raised hides (>3 m, 26%, usually for seated hiding during a single hunt, “high seat”). The proportions of types of stands used for drive hunts differed strongly within the diverse regions (Figure 6, chi^2^ goodness of fit test: df = 5, *p* < 0.001, drive hunt stand: *x*^2^ = 553.0; ground level: *x*^2^ = 331.8; raised hide: *x*^2^ = 152.4). Especially in those regions with low wild boar densities (N north, NW northwest, SW southwest, compare Figure 1 for a geographical overview), there seemed to be no tradition for drive hunt stands. Hunters did use raised hides quite a lot (existing structures) or just shot standing on ground level during small drive hunts within, e.g., fields, thickets or copses.

#### 3.2.5. Do You Have Any Difficulties in Conducting Drive Hunts? Which Are These?

(N = 2357, database: WTE 2012, singular query in 2012, predetermined answers)

62% (of those answering) had no difficulties in conducting drive hunts. In those regions with low wild boar densities, the effort seemed to be too high (24%). A small proportion (11%) had problems with adverse conditions for shooting due to dense vegetation. A total of 7% of the tenants had some problems selling the high amount of hunting bags. Only 6% had problems getting good dogs and shooters. All other reasons made up less than 5% (organisation 4%, building drive hunt stands 3%, hygienic reasons (cooling, water) 2%, trailing dogs 1%).

## 4. Discussion

The WTE does not only provide long-time and area-wide data on hunting bags and wildlife population estimates but also allows with its high rate of participation for insights into human habits. Not only was our data informative and enlightening on regional hunting methods, but we were also able to gain insight into hunters’ abilities, aims and opinions. The variety of different landscapes and different regions within Lower Saxony, as well as the notable participation quota in our study, would give us the ability to transfer our findings with other regions in Germany or even other countries in comparison (well knowing that there are few similar investigations). It would be worthwhile to conduct similar investigations on a broader basis on a European level and ask for changes in hunting attitudes especially in regions where African swine fever occurred.

In general, the rate of hunters answering more specific questions on how hunting was conducted (compare Figure 6) was much lower than the proportion of answers on more general hunting questions (see, e.g., Figure 2) (compare also [27]). This may mainly base on the occurrence of wild boar in Lower Saxony: hunters without resident wild boar occurrences cannot answer some of these specific questions generally [27]. Thus, the proportion of hunting grounds conducting wild boar drive hunts could be fostered by more conjoint drive hunts (several hunting grounds with transient wild boar work together). Additionally, possibly this may show some deficiencies in knowledge on how to hunt wild boar sufficiently, and perhaps, also reveals an existing proportion of inexpertness perhaps of the obstinacy of hunters (diminishing speed of innovations). Hunters should be skilled and trained furthermore in all aspects of wild boar hunting (a necessity for regulation, wild boar biology, possibilities for different hunting methods such as targeted single hunt or specific drive hunts especially in low-density areas) to ensure sufficient hunting performance, especially in regions with spreading populations.

### 4.1. Hunting Methods and Hunting Bags

The proportion of wild boar shot during drive hunts increased–especially on conjoint drive hunts–in the last two decades. Slight fluctuations might be due to due to weather conditions and population dimensions [76]. Collective hunting and drive hunts were promoted in German hunting magazines over several years as it is assumed to be most efficient [1], measured in man-hours, see [57,58,77,81]. Thus, it can account for more sufficient hunting methods, even on private hunting grounds. This shows that hunters might be on the right track. However, the proportions of hunting methods within hunting yields still differ regionally depending on hunting abilities, landscape structures, requirements, wild boar (occurrence, density, first appearance…), and attitudes [49,51,57,58]. Still, the most important method is hunting at bait (about one-third of the hunting bag in Lower Saxony) with the highest proportion in the western parts of Lower Saxony (compare Figure 3). However, the proportion of hunting grounds where baiting on wild boar is conducted is highest in eastern regions (compare Figure 4). This shows that the higher the wild boar population is, even on private hunting grounds where drive hunts occur, the more baiting is carried out. On the other hand, in some regions, the percentages of hunting grounds where baiting and/or drive hunts are conducted do not relate to wild boar population densities (compare also [26]). Perhaps other factors such as traditions, reservations against change, or a disparity in innovations to prevent wild boar from spreading in different regions, may lead to variation in hunting methods, attitudes, and behaviors [1,23].

Baiting is still a very important hunting method and is very efficient [51], measured in man-hours, see [57,58]. Until awareness is raised on other hunting methods, baiting will continue to be necessary for targeted and selective hunting [50,58] and will remain as one of the main tools in less forested regions with transient populations. Additionally, the quality of meat will usually be better under single hunt conditions [82]. Nevertheless, in order to prevent additional reproduction, it is important that wild boar are baited (small amounts of food) and not fed [34,83].

Astonishingly, only half of the hunting grounds in areas with resident wild boar conduct drive hunts. In order to regulate the wild boar population, this proportion has to be raised, especially by conjoint drive hunts to incorporate more hunting grounds and raise sufficient hunting performance [1,57,58]. Wild boar plays an important role in the conduction of drive hunts, as more than half of hunting ground tenants see wild boar as the main aim. In one-fifth of hunting grounds, wild boar is an incidental species (“bycatch”) on drive hunts. The majority (79%) of hunting ground tenants also regulate roe deer by drive hunts as secondary “aim-species” (which will give fewer disturbances compared to frequent additional single hunting days). The other species of hoofed game have more regional importance due to their distribution. We can also see that other species are important aims in drive hunts. Therefore, this hunting method is a good tool for the management of hoofed game. Conversely, the interest in additional species may interfere with the efficiency of wild boar management by private hunting.

The differing proportions of the conducted drive hunt methods are indicative of the hunters’ knowledge and attitudes, regional geography and vegetation, as well as wild boar densities. Well-known daytime resting habitats such as thickets and reed are regularly driven. Maize, intertillage and groves (small copses within fields) are driven mainly in the agriculturally-dominated areas due to their higher frequency. Other crops are driven simply because of actual wild boar occurrences. Slow beating and poking do not work for wild boar hunting and these methods are mainly used for mouflon or deer species (compare also [84,85,86,87]). During these hunts, wild boars are the “bycatch” species, as previously mentioned. Collective hides are able to hunt wild boar in agricultural areas and drive hunts in forested regions. Dogs are necessary for wild boar hunting [54,85,86,87] and as such, wild boars are rarely hunted without dogs.

Only a few hunters are having problems conducting drive hunts. These problems are minor and can be helped by conducting conjoint drives hunts, promoting cooperation, being trained by professionals, promoting infrastructure as well as sales, and distribution of game. Additionally, institutionalised professional wildlife managers as assisting organisers and motivating consultants could be instated. In agricultural areas drive hunts are more difficult to conduct. A serious proportion (11%) had problems with lack of opportunity for shooting due to dense vegetation. In hunting grounds with dense vegetation, very specific hunts and well-trained shooters are needed. However, conduction of drive hunts means a high effort. Thus, in regions with very low wild boar densities drive hunts will never make much sense.

For regions with low wild boar densities (N north, NW northwest, SW southwest, see Figure 1) there seems to be no tradition for drive hunt stands and these hunters use existing structures (raised hides) regularly, or just shoot on the ground level during drive hunts. Transportable stands could help increase the proportion of highly effective drive hunt stands (better than raised hides for the moving game) and ensure security (better than ground level stands).

Paying guests are uncommon in private hunting grounds. For private hunters, hunting is not their main source of income and they hunt with co-hunters and neighbours to regulate populations and for recreation. Perhaps the proportion of invited neighbours could be increased to promote cooperation.

### 4.2. How to Hunt? (With a Discussion on Efficiency Based on Literature Comparison)

In Lower Saxony, as in all other Federal States of Germany, the hunting bags are still increasing (see Appendix A) with some stable periods. It appears that private hunters with the help of science-based management commendations may be able to regulate wild boar populations on high-density levels in regions with high proportions of forested areas. However, some small changes in population, reproduction or environmental conditions may foster an increase anew. Spreading or recovering populations [27] seem to be hardly manageable by private hunting solely, especially in regions with low proportions of forested areas.

Drive hunts are known to be most efficient [1,56,57] but need to be conducted more frequently (compare also Appendix A in combination with Figure 2). Sustainability seems to be one of the main aims of the majority of private hunters (compare [27]), and thus, at least some hunters do take too much care of wild boar (this seems also to be similar for public forestry officers, compare also [27]). Thus, population regulation is hardly achievable [2]. The harvest rate of piglets, specifically, seems to be insufficient and could be raised by intensified and enhanced drive hunts [1]. The bias between reproductive [31,33,34] and harvest rates [1,88] needs to be counteracted with sufficient hunting performance. Additionally, single hunts within fields play an important role. Hereby hunters try to minimise damages [58], although they might risk shooting mothers (compare [49,89]). Besides early piglet hunting in agricultural fields, regulative hunting has mainly to be conducted in woodland during winter and is a precondition for regulation and prevention of damages in agricultural fields [1,44,49,51,58,66,89].

Intensifying conjoint drive hunts and small drives or battues (e.g., including more hunting ground, larger areas; increasing hunting pressure by more beaters, dogs, hunters; conducting hunts repeatedly; cooperating with hunters of neighbouring hunting grounds), especially for wild boar, may increase efficiency (in man-hours and sufficient hunting performance), especially in regions with high proportions of forested area, but should also be fostered in agricultural dominated regions (compare [1]). As most hunters do not see problems in conducting drive hunts this should be possible, although challenging, due to favoured sustainability instead of needed reduction [1,2,27]. This might be one of the biggest challenges within wild boar management: not the ability of hunters but the willingness to change systems and to reduce populations (compare [26,27]). If drive hunts are not practicable, and thus, high numbers of piglets are not reachable by private hunting–due to several factors (e.g., missing time, infrastructure, wild boar densities, willingness–especially in agricultural areas; [2,12,27])–intensified single hunt on adult females, assisted by professional hunters (wildlife managers), may be a reasonable solution [1,27,88].

Private (and also recreational) hunters contribute a significant portion to the wild boar harvest necessary for a population regulation [1,2,13]. However, in order to reduce populations (compare Appendix A) and prohibit epidemics (such as African swine fever ASF; lower densities have a lower risk for disease infection and especially for epidemic spread, compare [90,91]) more action is needed [1,2]. Furthermore, due to factors such as the willingness and abilities of individual hunters [27] additional efforts may be necessary. Thus, additionally professional hunters as employees (well known that this is a costly aim) of regional hunting associations or local/regional administrations, may serve as supportive wildlife managers. These could additionally train hunters, promote private hunters’ awareness of problems connected to wild boar (and wildlife) management, support cooperation of neighbouring hunters, and conduct additional hunt and other management measures (e.g., trapping, compare also [1,2,12,27]).

## 5. Conclusions

Much is written about wild boar hunting, but little is known about how hunters do. With our data, we gain a little insight into how hunting and wildlife management is conducted. Private hunting on wild boar is carried out in various ways and seems to be conducted in a mainly professional manner (at least in Lower Saxony’s hunting ground system with duty for regulation, custody (“Pflicht zur Hege”) and personal responsibility). Hunting methods strongly depend on regional conditions such as geography, vegetation and wild boar densities, as well as on hunters’ traditions and attitudes (compare [27]). Hunter’s behaviours and attitudes do change slowly. These changes could be fostered, and more sufficient hunting could be achieved by training and skill enhancement. However, opinion research is essential (compare also [23,27]). Further data on hunters’ opinions and attitudes are needed. For improving the management of game species, proper knowledge from permanent monitoring of both, animals’ behavior and population development, as well as human attitudes, is needed on a broad global scale.

## Figures and Tables

**Figure 1 animals-11-02658-f001:**
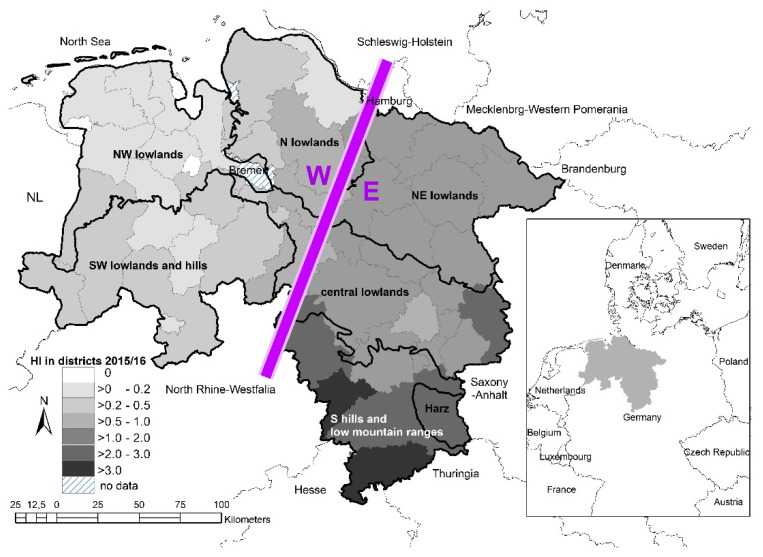
Wild boar hunting indices (HI = shot wild boar/km^2^ of hunting ground, hunting season 2014/15, official hunting statistics) for every administrative district in the Federal State of Lower Saxony, Germany, indicating wild boar abundances. Additionally shown the main geographic regions (broad black lines) N = North, S = South, E = East, W = West, HI = hunting index = shot wild boar/km^2^, NL = The Netherlands. Figure adapted from Keuling et al., 2016.

**Figure 2 animals-11-02658-f002:**
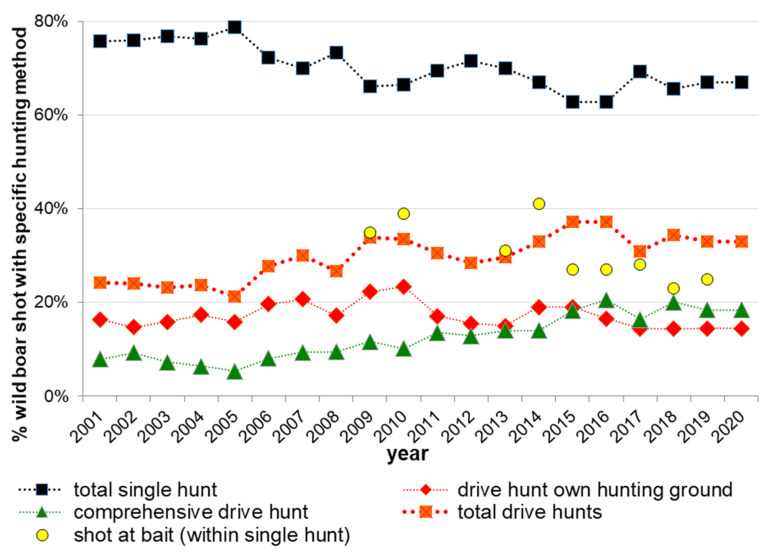
Development of proportions of the hunting methods single and drive hunt (shot animals as % of total hunting bag per hunting method) as a mean for Lower Saxony from 2001–2019 (2001 means hunting year 2001/02: 1 April–31 March). N = min 7286, max 8255, database: WTE 2002–2020. Bait was only surveyed in the shown years. Data only from WTE, as not available from official hunting bag data. Compare also Figure 3.

**Figure 3 animals-11-02658-f003:**
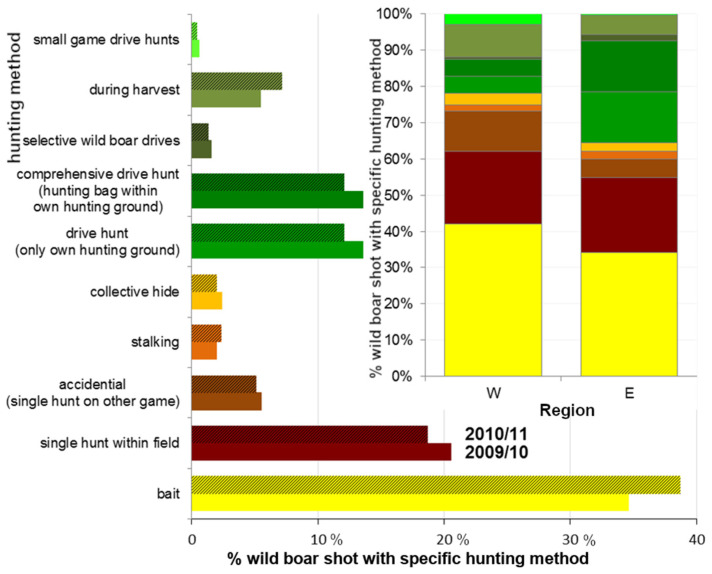
Percentages within hunting bag, wild boar shot with different hunting methods/strategies (predetermined answers). large graph: proportions total in Lower Saxony within the years 2009/10 (lower bar, N = 8106) and 2010/11 (upper bar, hatched colour, N = 8023), database: WTE 2010 + 2011; small graph: proportions in low (west = W, districts’ hunting index HI < 1 shot/km^2^) and high (east = E, HI > 1) density regions (2009/10), database: WTE 2010.

**Figure 4 animals-11-02658-f004:**
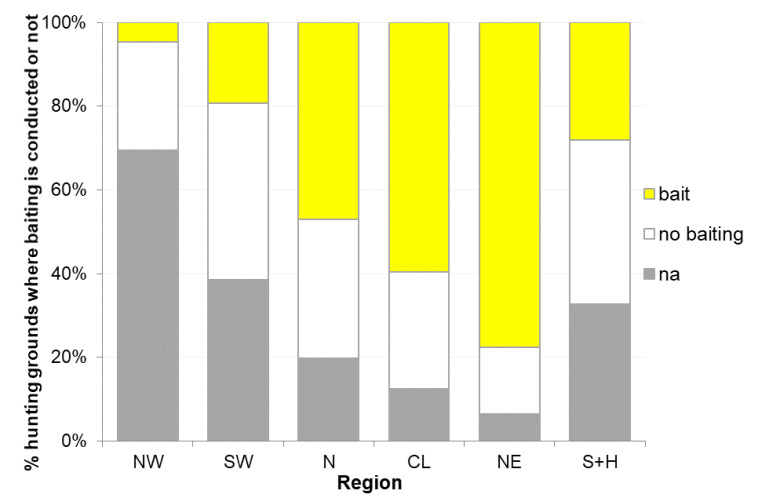
Percentages of hunting grounds with or without baiting for wild boar, N = North, S = South, W = West, E = East, CL = central Lowlands, H = Harz mountains, na = not answered. N = 7429, database: WTE 2012, singular query in 2012.

**Figure 5 animals-11-02658-f005:**
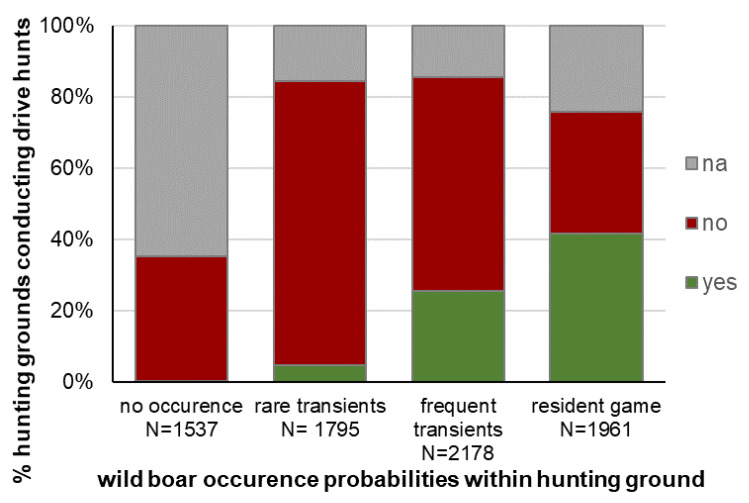
Percentages of hunting grounds conducting drive hunts on wild boar within different wild boar occurrence probabilities, na = not answered. N = 7907, database: WTE 2012, singular query in 2012.

**Figure 6 animals-11-02658-f006:**
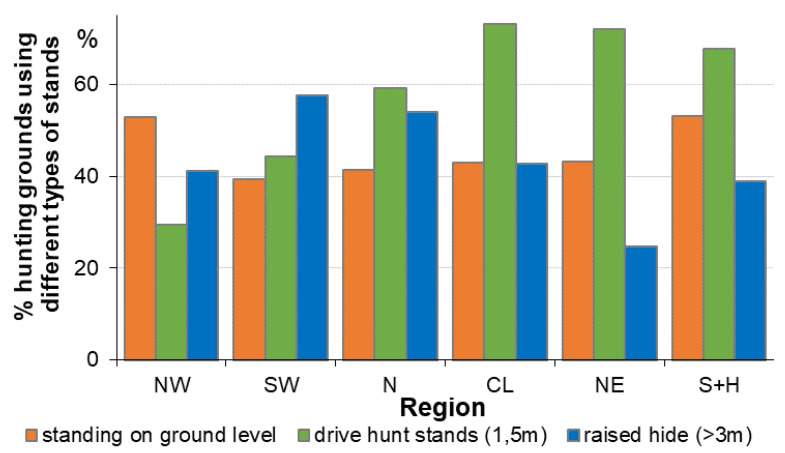
Percentages of hunting grounds using different types of stands within wild boar drive hunts. N = North, S = South, W = West, E = East, CL = central Lowlands, H = Harz mountains. N = 2119, database: WTE 2012, singular query in 2012.

**Table 1 animals-11-02658-t001:** Linear models explaining trends of proportions of the hunting methods single and drive hunt (shot animals as % of total hunting bag per hunting method).

Hunting Method	Variable	Estimate	SE	*p*	R^2^	F
Total single hunt	Intercept	13.6851	2.3321	1.48 × 10^−5^	0.6326	(1.18) = 30.99
	Year	−0.0065	0.0012	2.77 × 10^−0.5^		
Shot at bait (within single hunt)	Intercept	28.8145	8.9892	0.0150	0.5896	(1.7) = 10.06
	Year	−0.0142	0.0045	0.0157		
Total drive hunts	Intercept	−12.6370	2.2829	2.96 × 10^−5^	0.6407	(1.18) = 32.1
	Year	0.0064	0.0011	2.25 × 10^−5^		
Drive hunt in own hunting ground	Intercept	2.4215	2.0680	0.257	0.0617	(1.18) = 1.184
	Year	−0.0011	0.0010	0.291		
Comprehensive drive hunts	Intercept	−15.30	1.533	9.20 × 10^−9^	0.8491	(1.18) = 101.2
	Year	7.671 × 10^−3^	7.624 × 10^−4^	8.12 × 10^−9^		

**Table 2 animals-11-02658-t002:** The proportions of hunting methods within hunting statistics differ based on six regions in the years 2010 and 2011. Results of Chi^2^ goodness of fit test.

Hunting Method	df	*p*	*x^2^*
Single hunt at bait	5	<0.001	1492.0
Single hut within fields	5	<0.001	830.0
Accidential at hunt for other game from hide	5	<0.001	125.3
stalking	5	<0.001	115.3
Collective hide	5	<0.001	113.9
Drive hunt within own hunting ground	5	<0.001	1317.5
Comprehensive drive hunt	5	<0.001	1179.8
Selective wild boar drives	5	<0.001	144.7
During harvest	5	<0.001	189.8
Small game drive hunt	5	<0.001	364.2

## Data Availability

A file with prepared data will be available on the ResearchGate profile of the corresponding author (O.K. https://www.researchgate.net/profile/Oliver-Keuling (accessed on 6 September 2021). The raw data used herein may be made available upon reasonable requests to the Institute for Terrestrial and Aquatic Wildlife Research ITAW, University of Veterinary Medicine Hannover due to privacy protection solely by contacting the corresponding author (O.K.).

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
