# Peer review of "How Do Hunters Hunt Wild Boar? Survey on Wild Boar Hunting Methods in the Federal State of Lower Saxony"

_animals, 2021, doi:10.3390/ani11092658_

Round 1

Reviewer 1 Report

All the suggested revisions have been implemented or discussed in the revised version and rebuttal letter.

Author Response

Thank you for your efforts and help!

Reviewer 2 Report

In this new version of the manuscript, Keuling et al. solved the main concerns I had about the previous version. I consider the incorporation of new data and statistical analyses as the main improvements of the new submission.

The only remaining concern is regarding the conclusions of the manuscript (comment 4). Readers could expect the authors’ conclusions to answer the question in the title and address the objectives stated in the Summary, Abstract and at the end of the Introduction. In the first paragraph of the Conclusions section there is only one sentence related to their results, but the inclusion of the reference [27] can make readers doubt whether this information should be in the Discussion section.

Additional comments.

- Authors use sentences in present and past tense to refer to their results. Authors should decide which verb tense to use throughout the manuscript. The past tense is commonly used in scientific papers to refer to the authors’ results. However, after considering the title of the manuscript and the inclusion of recent data, doubts might arise.

- L198. Regression analyses were not used to “test” the development of proportions. Regressions analyses appear to be used to determine whether the slopes of tendencies were negative or positive (see e.g. L217, L223, L225 or L226).

- L219. The value of the X2 statistic is missing.

- L248.  “this did not differ between regions”, but the Kruskal-Wallis test show a p value lower than 0.05.

I encourage authors to revise all the statistical analyses and the statistical results they present in the manuscript.

Finally, I detected a typo: “startegies” in L26.

Author Response

R2: The only remaining concern is regarding the conclusions of the manuscript (comment 4). Readers could expect the authors’ conclusions to answer the question in the title and address the objectives stated in the Summary, Abstract and at the end of the Introduction. In the first paragraph of the Conclusions section there is only one sentence related to their results, but the inclusion of the reference [27] can make readers doubt whether this information should be in the Discussion section.

A: You are right. After reading it again and a again we shifted the last paragaraph to the discussion section.

Now it looks:

"Private (and also recreational) hunters contribute a significant portion to the wild boar harvest necessary for a population regulation [1,2,13]. However, in order to reduce populations (compare Supplemental Material S5) and prohibit epidemics [like african swine fever ASF; lower densities have a lower risk for disease infection and especially for epidemic spread, compare 90,91] more action is needed [1,2]. Furthermore, due to factors like willingness and abilities of individual hunters [27] additional efforts may be neces-sary. Thus, additionally professional hunters as employees (well knowing, that this is a costly aim) of regional hunting associations or local/regional administrations, may serve as supportive wildlife managers. These could additionally train hunters, promote private hunters’ awareness of problems connected to wild boar (and wildlife) management, sup-port cooperation of neighbouring hunters, and conduct additional hunt and other man-agement measures [e.g. trapping, compare also 1,2,12,27].

  1. Conclusions

Many is written about wild boar hunting, but little is known about how hunters do. With our data we get a little insight in how hunting and wildlife management is con-ducted. Private hunting on wild boar is done in various ways and seems to be conducted in a mainly professional manner (at least in Lower Saxony’s hunting ground system with duty for regulation, custody (“Pflicht zur Hege”) and personal responsibility). Hunting methods strongly depend on regional conditions like geography, vegetation and wild boar densities, as well as on hunters’ traditions and attitudes [compare 27]. Hunter’s behav-iours and attitudes do change slowly. These changes could be fostered and more sufficient hunting could be achieved by training and skill enhancement. However, opinion research is essential [compare also 23,27]. Further data on hunters’ opinion and attitudes are needed. For improving the management of game species, a proper knowledge from per-manent monitoring of both, animals behavior and population development as well as human attitudes is needed on a broad global scale."

R2: Additional comments.

- Authors use sentences in present and past tense to refer to their results. Authors should decide which verb tense to use throughout the manuscript. The past tense is commonly used in scientific papers to refer to the authors’ results. However, after considering the title of the manuscript and the inclusion of recent data, doubts might arise.

A: Thank you for that hint, we checked for that and found some sentences, where we changed the tense into past (esp. in results section). However, in many cases it had to stay in present, as it was belonging ongoing situations (Study area, description of survey etc.)

R2: - L198. Regression analyses were not used to “test” the development of proportions. Regressions analyses appear to be used to determine whether the slopes of tendencies were negative or positive (see e.g. L217, L223, L225 or L226).

A: Of course, you are right! Thank you for that hint. We changed as: "To characterize the development of proportions over time we calculated linear regressions determining whether the slopes of tendencies were negative or positive."

R2:- L219. The value of the X2 statistic is missing.

A: Forgot, as these were 10 x². we added them now!

R2:- L248.  “this did not differ between regions”, but the Kruskal-Wallis test show a p value lower than 0.05.

A: True. This was a mistake. To make this more clear we added: "Those hunting grounds with baiting stations had a mean of 2.6 baiting stations per hunting ground; this did differ between regions (W: 2.2 bs/hg, E: 2.7 bs/hg, see Figure 1; Kruskal-Wallis-test: x² = -4541.5, df = 5, p < 0.001)."

I encourage authors to revise all the statistical analyses and the statistical results they present in the manuscript.

A: done

Finally, I detected a typo: “startegies” in L26.

A: Changed!

Reviewer 3 Report

Overall this is a reasonably interesting piece of work that allows some quantification of the wild boar hunting in Lower Saxony and an idea of methods and issues that can be explored further in future.

The work is reasonably well written but in some places the translations do not fully work and I would suggest further edit to ensure the meaning is clear.  In particular

Lines 51 - 55 do not read well

Lines 67-71 need a bit more clarity to explain the terms and the writing could be more concise here.

I am not sure the questions asked in the questionnaire allow you to address the issue of "are hunters willing and able to regulate a population sufficiently". Can you clarify how this has been assessed via the questionnaire or edit the aims of the research.

Additionally you have addressed what difficulties have been encountered but not specifically what management tools can be used and accepted. 

Therefore the aims on lines 102-104 need to be clarified or justified.

What prior assumptions have been tested to ensure that the linear regression is a suitable test?

The results would be better laid out if each question was set out as a subsection as they are difficult to follow currently.

You have not reported any significance value for the regressions and the regression slopes appear to be very weak-line 218, 225 and 228.

Chi squared results on line 219-224 would be better set out in a table

All graphs need to have the axes labelled.

Please review the writing in the discussion for example line 428 - more common and line 433 hardly to - don't make sense.

.

Author Response

This manuscript is a resubmission of an earlier submission. The following is a list of the peer review reports and author responses from that submission.

Round 1

Reviewer 1 Report

The paper “How do hunters hunt wild boar?” by Oliver Keuling, Egbert Strauß and Ursula Siebert is an interesting paper face the topic of wild boar with an approach typical of human dimension but with implication of wildlife management.

The paper il well written and comprehensively deepen some different aspects of the problems arising from increase of WB populations. The language is quite simple and easy to understand, although English editing is not needed, some concepts, unfamiliar for the reader could be more clearly explained.

The paper is sound in the field of study and can provide an increase in scientific knowledge and approach.

Some detailed comments below.

Improvement could be made both in simple summary and abstract that should meake the topic understandable before reading the whole text.

L 14-16 an 30-31 “as hunters have only few difficulties in conducting conjoint hunts on wild boar” for me is unclear

L 18-21 and 32-35 the period is unclear.

L 29 geographical conditions ?

L 37 key words: wild boar could be added?

L 54 food conditions?

Throughout the introduction and material and methods chapter any reference to unmanaged areas by the Hunters (hunting ban) is not reported. Are not present in the area? The role of unmanaged areas is compatible with the assertion al L 46-47?

L 177 However, actual hunting is conducted in terms of the tenants' targets – is unclear

L 192 participating hunting grounds in 2002-15, see fig 2 seems 2001-2014

Figure 3: is unclear, the horizontal axis is %? The figure is a very good synthesis but is very difficult to understand . The figure should be easy to understand without reading the text

Figure 4: the legend could be a little bit larger

Figure 5: yes/no/na are not visible  (especially black on brown), the column are very large, may be using the same scheme of fig 4 is more clear?

L 280 types of stands within wild boar – I have some concern about the term battue and is unclear the ground level, it means lying on the ground?

L 395 please check the literature names instead of numbers!

L 409 - Intensifying conjoint drive hunts and small battues, especially is unclear by the technical point of view.

About the supplementary material I think that the explanation of techniques is very useful, the graphs are too little to be clearly understandable.

As some details of wildlife management are underconsidered in this paper the Authors shoyld stress the "human dimension" 

Reviewer 2 Report

In this manuscript, Keauling et al. show the results of enquiries on hunting methods carried out in Lower Saxony (Germany). They gathered a great deal of information with these enquiries. In addition to the amount of information they collected, I find a valuable result in the manuscript:

Authors mention that wild boar populations are increasing and dispersing into new areas. This population increase causes problems, and it is a relevant threat. Harvest rates appear to be insufficient in many parts of the wild boar range (Introduction, L40-45). On the other hand, authors mention that drive hunts are assumed to be the most efficient hunting method (eg. Discussion, L318-319). They found that the percentage of wild boar shot with drive hunts tended to be around 30% in the period 2001-2014, and this percentage tended to increase in the period 2005-2014 (Figure 2). Authors conclude that hunters might be on the right track (eg. L321), but the proportion of drive hunts should be fostered (eg. L32, L305-306, L343-345).

Despite the valuable aspects of the manuscript, there are drawbacks that may prevent its publication in the present form.

Comments.  

1) The first main comment is related to the title: “How do hunters hunt wild boar?”. When I first read the title, I was expecting things that I finally did not find. On the one hand, I expected the analysis of large-scale hunting methodologies. However, I found the hunting methodologies that are carried out in Lower Saxony. In other parts of the word, even in other regions of Europe, hunters hunt wild boar in different ways. This might be solved by modifying the title. On the other hand, the title is expressed in the present tense. However, they analyzed the data from a period between 2002 and 2015. Even most of results were obtained from inquiries conducted in 2012. I guess the singular query conducted in 2012 cannot be repeated. However, is there any reason not to continue the collection of WTE data? Figure 2 is very interesting, but it is frustrating not to find the trends of the last few years. Probably data from 2020 are not available, but how were the trends until 2019? I acknowledge that data are not always available for researchers, but the title of the manuscript might need the data of the last few years. In the manuscript, there is a sentence that illustrate my concern: “The proportion of wild boar shot during drive hunts increased - especially on con joint drive hunts – IN THE LAST TEN YEARS with slight fluctuations due to weather conditions and population dimensions.” (L316-318).  

2) Authors show their results with descriptive statistics (mainly percentages). They did not use any statistical test to obtain results. The lack of statistical tests prevents drawing conclusions from the data collected. Authors write statements that can not be derived from the results they provide. For instance: “Since 2005, the proportion of wild boar shot on drive hunts has increased considerably, specifically where conjoint drive hunts are promoted” (L205-206); “The proportion of hunting grounds where drive hunts are conducted differs due to the occurrence, frequency and density of wild boar” (L235-236); “However, the proportion of hunting grounds where baiting on wild boar is conducted is highest in east ern regions” (L326-328); “This shows that the higher the wild boar population is, even on private hunting grounds where drive hunts occur, the more baiting is done.” (L328-329); “On the other hand, in some regions, the percentages of hunting grounds where baiting and/or drive hunts are conducted do not relate to wild boar population densities” (L329-331). These sentences are examples that illustrate my concern. All these statements must be based on statistical tests that assess null hypotheses.

3) L13 and L26. In this work, authors do not deal with hunting efficiency. Here, they show the percentages of wild boars shot by each hunting method. Hunting efficiency has been previously assessed in other studies. A relevant statement in the manuscript is that drive hunt is an efficient method to hunt wild boar. But this statement is not based on the results of this work. The usage of “hunting efficiency” in the simple summary and the abstract might confuse readers.   

4) From my point of view, the Conclusions section needs to be improved. The first paragraph includes statements that are not conclusions of the work. Regarding the second paragraph, authors should strive to show what this work adds to what is already known (see L64-79 in Introduction section).

5) L203. “Hunting at bait IS the most common hunting method in Lower Saxony”. This statement is based on the results of two seasons, ten years ago. Do authors have data to assess whether this result was maintained across years?

6) L203-205. To which extend does the proportion of hunting methods differ between regions? Please provide statistics and test the null hypothesis that the proportion of hunting methods does not differ between regions. See comment 2.

7) L223-225. Authors should provide the result of a test assessing whether baiting was conducted more frequently with higher wild boar densities. See comment 2.

8) L235-236. “The proportion of hunting grounds where drive hunts are conducted differs due to the occurrence, frequency and density of wild boar”. Results in present tense with data from 2012 and without a statistical test that support this affirmation.

9) L267. Probably the percentage between regions were not identical. Please provide statistics and test the null hypothesis that the percentage of hunting party does not differ between regions. The comparison between comment 6 and 9 illustrate the issue provided in the comment 2.

10) L273-274. “The proportions of types of stands used for drive hunts differ strongly within the diverse regions”. Results in present tense with data from 2012 and without a statistical test that support this affirmation.

11) L301-303. “In general, the rate of hunters answering more specific questions on how hunting was conducted was much lower than the proportion of answers on more general hunting questions”. Is there a result that support this information?

12) L317-318. “…due to weather conditions and population dimensions”. Is there a result that support this information?

13) L358-359. “Maize, intertillage and groves (small copses within fields) are driven mainly in the agricultural dominated areas due to their higher frequency”. Results in present tense with data from 2012 and without a statistical test that support this affirmation.

Additional comments

L42 and L64-65. Wild boar densities and dispersal into new areas are not only economic problems. The consequences of wild boar densities are beyond of only an economic issue.

L50. “Additionally” instead of “Nevertheless”?

L53. Could be changed the expression “In common opinion…”?

L57-61. Authors highlight where studies are lacking without clear evidence. Unexpectedly, authors show a higher number of references in those items in which they detect lack of research.  

L156-157. More information should be provided regarding these types of data. Mainly: “own calculations based on official hunting bag statistics”.

L180-181. Perhaps, more information might be provided regarding the evaluation of the reliability of estimates.

Figure 2. Why does the sum of the proportion of wild boars shot in both types of drive hunts differ from the total drive hunts?

Figure 3, L214. “Percentage” instead of “Proportion”

L258-259. Please, provide concrete information instead of “specially high”, “regularly”, or “seldom”.

L275-276. Figure 6 instead of figure 1?

Reviewer 3 Report

Thank You very much for an opportunity to review this paper. I think it might be interesting only from local (regional) point of view. The paper does not present broad scope of the problem. In my opinion in its present form paper is not appropriate for Animals  and should be submitted to local journal from Germany. It might be more interesting to readers if Authors collect and add data from different parts of Germany or from other EU countries and compare them. I am also interested what is happening currently in this hunting grounds during African Swine Fever epidemic. What has been changed in the last 6 years and what is current situation of wild boar population. 

I also do not understand how to interpret the figure 1? Do Authors think that high hunting bag reflects high population density? or it is a result of other factors like number of hunters, crop damages......etc.